# CircEHD2, CircNETO2 and CircEGLN3 as Diagnostic and Prognostic Biomarkers for Patients with Renal Cell Carcinoma

**DOI:** 10.3390/cancers13092177

**Published:** 2021-04-30

**Authors:** Lisa Frey, Niklas Klümper, Doris Schmidt, Glen Kristiansen, Marieta Toma, Manuel Ritter, Abdullah Alajati, Jörg Ellinger

**Affiliations:** 1Department of Urology, University Hospital Bonn, 53127 Bonn, Germany; s4lifrey@uni-bonn.de (L.F.); niklas.kluemper@ukbonn.de (N.K.); doris.schmidt@ukbonn.de (D.S.); mritter@ukbonn.de (M.R.); abdullah.alajati@ukbonn.de (A.A.); 2Institute of Pathology, University Hospital Bonn, 53127 Bonn, Germany; glen.kristiansen@ukbonn.de (G.K.); marieta.toma@ukbonn.de (M.T.)

**Keywords:** circular RNA, renal cell carcinoma, biomarker, circEHD2, circNETO2, circEGLN3

## Abstract

**Simple Summary:**

Circular RNA (circRNA) plays an important role in cancer, but little is known about its role in clear cell renal cell carcinoma (ccRCC). The study was designed to analyze the role of circRNAs in ccRCC. We show that circEHD2, circENGLN3, and circNETO2 are upregulated in ccRCC compared with non-malignant renal tissue. Increased circEHD2 levels were significant and independent predictors of progression-free and cancer-specific survival of ccRCC patients. Thus, the analysis of circRNAs may be of diagnostic and prognostic relevance in patients with ccRCC.

**Abstract:**

Background: Circular RNA (circRNA) plays an important role in the carcinogenesis of various tumors. It is assumed that circRNAs have a high tissue and tumor specificity, thus they are discussed as cancer biomarkers. The knowledge about circRNAs in clear cell renal carcinoma (ccRCC) is limited so far, and thus we studied the expression profile of seven circRNAs (circCOL5A1, circEHD2, circEDEM2, circEGNL3, circNETO2, circSCARB1, circSOD2) in a cohort of ccRCC patients. Methods: Fresh-frozen normal and cancerous tissues were prospectively collected from patients with ccRCC undergoing partial/radical nephrectomy. Total RNA was isolated from 121 ccRCC and 91 normal renal tissues, and the circRNA expression profile was determined using quantitative real-time PCR. Results: circEHD2, circENGLN3, and circNETO2 were upregulated in ccRCC compared with non-malignant renal tissue. circENGLN3 expression was highly discriminative between normal and cancerous tissue. None of the circRNAs was correlated with clinicopathological parameters. High circEHD2 and low circNETO2 levels were an independent predictor of a shortened progression-free survival, cancer-specific survival, and overall survival in patients with ccRCC undergoing nephrectomy. Conclusions: The analysis of circRNAs may provide diagnostic and prognostic information. Thus, circRNAs could help to optimize the individual treatment and ultimately improve ccRCC patients’ survival.

## 1. Introduction

Renal cell carcinoma (RCC) is one of the most frequent cancers with substantial mortality. Early-stage RCC may be cured by surgery, however, the prognosis of patients with metastasized RCC is poor. The armament of anticancer therapies has increased in the past years: immune checkpoint inhibitors, tyrosine kinase-inhibitors, VEGF-antibody, and mTOR inhibitors. These improved survival, but a cure is not achievable [1,2]. A better understanding of the molecular mechanisms underlying RCC could help to develop new therapeutic strategies. Furthermore, molecular biomarkers could indicate individual prognosis, detect cancer recurrence/progression early, and allow a more individualized therapy.

Circular RNA (circRNA) was discovered in the 1970s and was initially considered as a functionless RNA splicing product [3]. circRNA has a loop structure with covalent bonds of head and tail. circRNAs lack a 3′ end which protects them against exonuclease digestion and contributes to their higher stability compared with mRNA [4]. Circular and linear RNAs are derived from precursor mRNAs, and different splicing modes have been discovered. According to the location of splice junction in the genome, circRNAs are classified as exonic circRNA, circular intronic RNA, exonic–intronic circRNA, or tRNA intronic circRNA. Exonic circRNAs are formed by either single or several exons and are the most common circRNA [5]. Knowledge about the biological functions of circRNAs is still limited. However, circRNAs have various functions. They regulate gene expression on the transcriptional and the translational level, interact with proteins, act as miRNA sponges, translate proteins or peptides, modify rRNA processing, and generate pseudogenes [3]. Technological advances lead to the discovery of numerous circRNAs. Importantly, the cellular circRNA expression profile has a high tumor and tissue specificity [6,7], thus circRNAs have potential as a new cancer biomarker.

The knowledge of the expression of circRNAs in RCC is limited. In a recent publication, small-scaled gene expression analyses demonstrated a RCC-specific circRNA expression profile which allowed discrimination of tumor from normal renal tissue [8,9,10,11]. We systematically reviewed the former studies employing microarray expression profiling to study circRNA in ccRCC to identify circRNA candidates for a validation study. The small-scaled microarray analyses reported upregulation of various circRNAs, however most of them were not validated in a large cohort. Among these circRNAs, circENGL3 [8,12], circCOL5A1 [9], circEHD2 [8], circNETO2 [8], circSCARB1 [8], circSOD2 [9], and circEDEM2 [9] were of particular interest as biomarkers due to distinct overexpression in RCC (e.g., >4-fold). The analysis of circEGLN3 in an enlarged cohort confirmed specific upregulation in clear cell renal cell carcinoma (ccRCC) tissue. Furthermore, circEGLN3 expression was associated with cancer-specific survival following nephrectomy [8]. The remaining circRNAs have not been validated by independent studies so far. Our study was therefore designed to study the expression of seven circRNAs (circENGL3, circCOL5A1, circEHD2, circNETO2, circSCARB1, circSOD2, circEDEM2) in a large cohort of ccRCC and normal renal tissues to increase the knowledge on the potential role of circRNAs as biomarkers for ccRCC.

## 2. Materials and Methods

### 2.1. Patients

Tissue samples were collected prospectively in the biobank of the Centre for Integrated Oncology at the University Hospital Bonn from patients who underwent radical or partial nephrectomy at the Department of Urology. The study was composed of two cohorts. First, a discovery cohort with 20 tissues of patients with clear cell renal cell carcinoma (ccRCC) and 10 normal tissues was studied. The ccRCC samples were selected to include each 10 patients with localized and metastatic ccRCC in order to allow identification of diagnostic and prognostic relevant circRNA biomarkers. Second, a validation cohort containing 101 ccRCC samples and 81 normal renal tissues was selected randomly from the biobank. All benign renal tissues were derived from nephrectomy specimens; the tissue was taken distantly from the tumor and histologically normal renal tissue. Fresh-frozen tissues were stored at −80 °C. All used tissues were re-examined by a uro-pathologist and classified following the 2009 WHO classification. The clinicopathological information of the patients is provided in Table 1. The study was approved by the Ethics Committee of the University Bonn (vote: 347/19).

### 2.2. RNA Isolation, RNase R Treatment, and cel-miR-39 Spike-in

The RNA isolation method was described in brief earlier [13]. Total RNA was isolated from 50 mg fresh-frozen renal tissue using the mirVana miRNA Isolation Kit (Ambion, Foster City, CA, USA) and treated with DNase (DNA-free Kit, Ambion). All procedures were performed according to the manufacturers’ recommendations. The RNA concentration was measured using a NanoDrop 2000 spectrophotometer (Thermo Scientific, Wilmington, DE, USA) and stored at −80 °C.

To remove linear RNA, we treated the samples with RNase R (Lucigen, Teddington, United Kingdom). The RNA was diluted to 500 ng in a total volume of 20 µL and the treatment was done according to the manufacturers’ instructions (RNase R 0.5 U/µL, RNase R buffer 2.0 µL, and RNase Inhibitor N8080119 2.0 µL) at 37 °C for 90 min.

Thereafter, the RNA was purified by a phenol/chloroform extraction and ethanol precipitation. cel-miR-39 (Qiagen, Hilden, Germany) (0.05 fmol/µL) was spiked-in. After ten minutes of centrifugation, the upper aqueous phase was transferred to a new tube. Sodium acetate (3 M) and absolute ethanol were added and incubated overnight at −80 °C. After washing with ethanol and centrifugation at 4 °C, the ethanol was removed and the RNA resuspended in water.

### 2.3. cDNA Synthesis and Quantitative Real-Time PCR

cDNA synthesis and quantitative real-time PCR have been described earlier [14]. In brief, cDNA synthesis was performed with 500 ng RNA using the PrimeScript reagent Kit (Takara, Saint-Germain-en Laye, France). The manufacturers’ protocol was used, with the exception of adding oligo(dT)-primer. It was incubated at 37 °C for 15 min and heated to 85 °C for five seconds. Quantitative real-time PCR was done using the Takara TB Green Premix Ex Taq II with 1 ng/µL RNA and 10 pmol/µL forward/reverse primer on the QuantStudio 5 Real-Time PCR System (Applied Biosystem by Thermo Fisher Scientific, Waltham, MA, USA) on 384-well plates (Thermo Fisher) in triplicates. No template controls, no reverse transcription controls, and genomic DNA controls were included and showed negative results. Relative circRNA expression levels were determined using the QuantStudio 3D AnalysisSuite Cloud Software (Applied Biosystems by Thermo Fisher Scientific); cel-miR-39 was used as the reference gene.

Various circRNAs have been identified as potential diagnostic biomarkers in small-scaled microarray experiments; among these circRNAs, circCOL5A1, circEHD2, circEDEM2, circNETO2, circSCARB1, and circSOD2 demonstrated a significant and at least 4-fold overexpression and were thus chosen for validation in our study. The primer sequences of circEGLN3 [6], circSOD2 [15], and circEDEM2 [16] were published earlier. Divergent circRNA-specific PCR primers were designed for circNETO2 (forward: 5′-AGT-GAT-TCG-AAT-GTG-GGC-AGA-3′; reverse: 5′-ATT-AAC-AAC-AGC-TCC-ACA-AAG-GA-3′), circSCARB1 (forward: 5′-AGG-TTC-AGT-TGA-CTT-CTG-GCA-3′; reverse: 5′-CTC-AGG-AGT-CAT-GAA-GGG-CG-3′), circEHD2 (forward: 5′-CTG-GTG-CGA-GCT-ACG-ACT-TC-3′; reverse: 5′-TCG-TCC-GAG-ATC-TCC-AGC-TT-3′), and circCOL5A1 (forward: 5′-CCA-AGG-ATG-CTC-CAG-GGA-TT-3′; reverse: 5′-GGC-CCC-CTT-CGG-ACT-TCT-3′) using CircInteractome [17]. The PCR efficiency was 103.2% for circEGLN3, 95.5% for circEHD2, and 95.9% for circNETO2. The PCR primer for cel-miR-39 (used as reference gene) was purchased from Qiagen (miScript Primer Assay, catalog no. 218300). See Appendix A for more information on the design of divergent PCR primers and the MIQE-report (Appendix A).

### 2.4. Statistics and Data Analysis

SPSS IBM Statistics 25.0 was used for statistical analysis; *p* < 0.05 (two-sided) was considered as statistically significant. The Mann-Whitney U test was used to compare the circRNA expression with clinicopathological parameters. To identify the prediction capacity receiver operating characteristic (ROC) curves with area under the curve (AUC), analyses were done. To determine the optimal cutoff for each circRNA for the prediction of survival, we used the Cutoff Finder algorithm [18]. Kaplan–Meier curves for circNETO2 and circEHD2 were then generated using SPSS. For survival analysis, Kaplan–Meier estimates and Cox proportional regression analysis (using the dichotomized circRNA variables) were applied.

## 3. Results

### 3.1. Discovery Cohort

We first studied the circRNA expression of seven circRNAs in a discovery cohort including 20 ccRCC and 10 normal renal tissues (NAT). The expression of circEGLN3 (*p* < 0.001), circEHD2 (*p* < 0.001), circNETO2 (*p* = 0.024), and circEDEM2 (*p* < 0.001) was significantly increased in ccRCC. The expression of circCOL5A1 (*p* = 0.307), circSCARB1 (*p* = 0.061), and circSOD2 (*p* = 0.055) was not significantly different in ccRCC and NAT; see Figure 1. Using ROC analysis, a sensitivity of 80% and a specificity of 100% for circEHD2; a sensitivity of 90% and a specificity of 80% for circEGLN3; a sensitivity of 70% and a specificity of 70% for circNETO2; and a sensitivity of 80% and a specificity of 100% for circEDEM2 was determined.

### 3.2. Validation Cohort

We next studied circEGLN3, circEHD2, and circNETO2 expression in a validation cohort consisting of an independent cohort of 101 patients with ccRCC and 81 NAT. As expected, circEGLN3, circEHD2, and circNETO2 (all *p* < 0.001) levels were increased in patients with ccRCC. circEDEM2 showed no significant difference in the expression level (*p* = 0.431) in the validation cohort. ROC analysis indicated that circEGLN3 allowed the most accurate discrimination of ccRCC and NAT (AUC = 0.879; sensitivity 87.1%, specificity 77.8%) followed by circEHD2 (AUC = 0.757; sensitivity 52.5%, specificity 85.2%) and circNETO2 (AUC = 0.705; sensitivity 59.4%; specificity 82.7%); see Figure 2.

We next analyzed the prognostic relevance of circRNA expression. None of the studied circRNAs were correlated with adverse clinicopathological parameters (i.e., TNM-stage, grade, gender). For survival analysis, we dichotomized the ccRCC using the Cutoff Finder [18] cohort into a high and a low expression group based on an optimized cut-off discriminating between survivors and deaths. High circEHD2 levels were associated with a significantly shortened progression-free survival (*p* = 0.002) and cancer-specific survival (*p* = 0.032). Low circNETO2 expression levels were predictive of a shortened progression-free survival (*p* = 0.001), cancer-specific survival (*p* = 0.001), and overall survival (*p* = 0.001) (see Figure 3). Furthermore, univariate and multivariate Cox regression analyses demonstrated that low circNETO2 and high circEHD2 were correlated with progression-free survival (see Table 2), overall-survival (see Table 3), and cancer-specific (see Table 4) survival. We decided to group the parameters and not to use continuous variables due to a lack of distant metastasis and a bigger and more representative cohort of each group. Thus, both circRNAs are predictors of patients’ outcome independent of clinicopathological parameters like tumor grade, pT-stage, and cM-stage.

### 3.3. Analysis of circRNA Interactions

We next analyzed potential interactions of circRNA (circEGLN3, circED2, circNETO2) with RNA-binding proteins. Using the CircInteractome [17] tool, we observed that multiple RNA-binding proteins have putative binding sites at the circRNA junction site but also the flanking region. The putative binding sites are displayed schematically in Figure 4. circRNAs may function as miRNA sponges. The CircInteractome tool revealed numerous target sites for specific miRNAs. The algorithm predicted up to two binding sites per miRNA in circEGLN3, circED2, circNETO2; see Figure 5.

## 4. Discussion

circRNA expression profiles provide a high tumor and tissue specificity [6,7]. circRNAs are highly resistant to RNase R digestion resulting in an increased half-life compared with other RNA molecules. They represent therefore a potential new cancer biomarker. It should be noted that a recent study questioned whether circRNAs have an increased stability compared with mRNA in tissue samples [19]. However, Rochow et al. [19] observed that circRNA expression is not altered in clinical samples with a high RNA integrity number (RIN > 6). We exclusively used fresh-frozen tissues in order to achieve a high RNA integrity, and therefore do not expect a bias in the circRNA expression. circRNA expression is also of interest for RCC research [20], although the knowledge of circRNAs in ccRCC is limited so far. We therefore studied the expression of seven circRNAs (circCOL5A1, circEHD2, circEGLN3, circEDEM2, circNETO2, circSCARB1, circSOD2) in a large cohort of ccRCC patients. Our study includes the largest number of normal and ccRCC tissues so far, and thus powerful statistical analyses are feasible.

One of the most promising circRNA ccRCC biomarkers is circEHD2. circEHD2 expression was significantly increased in ccRCC compared with normal renal tissue. Furthermore, high circEHD2 expression was an independent predictor of progression-free and cancer-specific survival in patients undergoing radical or partial nephrectomy. circEHD2 was identified by Franz et al. [8] using a microarray screening approach; the study included seven matched normal and ccRCC tissues. However, circEHD2 was not further validated. We therefore show for the first time that circEHD2 may be of diagnostic/prognostic relevance in patients with ccRCC. However, circEHD2 had a moderate diagnostic accuracy in our study (AUC = 0.757), and circEGLN3 may be better suited as a diagnostic biomarker due to an increased accuracy (AUC = 0.879). So far, circEHD2 has not been studied in other malignancies, and thus it is speculative whether this circRNA is specifically dysregulated in ccRCC.

circEGLN3 was distinctly upregulated in ccRCC tissue in our study; using analyses we observed that circEGLN3 allowed discrimination of ccRCC and normal renal tissue with an AUC of 0.879, thus the diagnostic sensitivity was 83.2% and the specificity was 80.2%. A former study [6] indicated an even more accurate diagnostic classification, with a sensitivity and specificity of 95%. However, the patient cohort in the study by Franz et al. [8] was smaller-sized (*n* = 82 non-metastatic and *n* = 17 metastatic ccRCC) and included a higher percentage of advanced ccRCC compared with our study (metastatic ccRCC in the validation cohort: *n* = 6; 5%). Significant overexpression was also observed by Lin and Cai [12]. Furthermore, Franz et al. observed low circEGLN3 expression levels as a poor prognostic parameter which correlated with survival following nephrectomy. In contrast, Lin and Cai [12] reported high circEGLN3 expression as a risk factor for shortened survival after nephrectomy. We did not observe any correlation with adverse parameters nor survival with circEGLN3. Knockdown of circEGLN3 resulted in an impaired proliferation, migration, and invasion, and facilitated apoptosis of RCC cells. It was shown that circEGLN3 acts as an oncogene through upregulating IRF7 via sponging miR-1299 in ccRCC [12].

Franz et al. [8] identified increased circNETO2 expression in the small-scaled microarray screening but did not validate this finding in a larger cohort. Our results confirm circNETO2 overexpression in ccRCC. Furthermore, survival analyses demonstrated that circNETO2 was a predictor of progression-free, cancer-specific and overall survival independent from clinicopathological parameters. The diagnostic relevance of circNETO2 may be limited as the diagnostic accuracy was moderate in our study (AUC = 0.705); circNETO2 has not been studied in RCC by other researchers so far. In lung cancer cells, circNETO2 expression was not different in normal and cancerous lung tissue [21].

The expression of circSCARB1 [8], circCOL5A1, circSOD2, and circEDEM2 [9] was reported as dysregulated in earlier studies, and we aimed to study these circRNAs in a larger cohort. In contrast to former studies (*n* = 99; *n* = 64), our study was enlarged, and the quantification technique (quantitative real-time PCR) had an increased accuracy compared with microarray analyses. Although we could not confirm a diagnostic relevance for any of these circRNA, they have been associated with oncogenic functions in cancer in former studies: circSCARB1 promoted RCC progression by sequestering miR-510-5p and indirectly up-regulating SDC3 expression [22], and circCOL5A1 acted as a miR-1224-5p sponge, thereby activating CREB1 expression and promoting cellular proliferation in bladder cancer [23]. circSOD2 expression enhanced cancer cell growth, cell migration, cell cycle progression, and in vivo tumor growth in hepatocellular cancer; circSOD2 inhibited miR-502-5p expression, thereby upregulating DNMT3A expression [24].

We finally addressed circRNA interactions with RNA-binding proteins and miRNAs. We observed that circEHD2, circEGLN3, and circNETO2 had putative binding sites for the RNA-binding protein EIF4A3 at the junction and 5′-flanking sites. EIF4A3 is implicated in cellular processes involving the alteration of RNA secondary structure, such as translation initiation, nuclear and mitochondrial splicing, and ribosome and spliceosome assembly. EIF4A3 binding sites are common in in the flanking region of circRNA junctions and may indicate a role of EIF4A3 in circRNA biogenesis [17]. In fact, EIF4A3 induced circMMP9 expression in glioblastoma [25] and circSEPT9 in breast cancer cells [26]. In addition, several putative miRNA target sites are located in circEHD2, circEGLN3, and circNETO2. Thus, these circRNAs may act as miRNA-sponge. We identified 15 (circEHD2), 19 (circNETO2), and 33 (circEGLN3) different miRNAs that had putative binding sites in the circRNAs, and thus these circRNAs may have complex functions on the miRNome of ccRCC.

Some limitations of our study should be acknowledged. The validation cohort included only few patients with distant metastases (*n* = 6) and no patients with lymph node metastases, and thus advanced stages are probably somewhat underrepresented in our cohort. However, cytoreductive nephrectomy is nowadays only indicated in a very small number of patients with metastatic disease, as the CARMENA [27] trial indicated improved outcome in patients undergoing sunitinib therapy without cytoreductive nephrectomy. The lack of metastatic cases in the validation cohort is on the other hand a strength of our study. The survival analyses indicate a prognostic value of circRNA expression in patients undergoing nephrectomy for localized RCC. Thus, the analysis of circRNA expression could aid the clinician to identify patients at increased risk of RCC recurrence after surgery with curative intention and thereby help to personalize therapy. Furthermore, these patients could be of special interest for adjuvant tumor therapy. As the number of patients with metastatic ccRCC was very small, we did not include risk groups (i.e., Heng or Motzer criteria) as a variable in the Cox regression analysis. We have chosen to validate only a limited number of circRNAs (*n* = 7); these circRNAs had in previous studies (circENGL3 [8,12], circCOL5A1 [9], circEHD2 [8], circNETO2 [8], circSCARB1 [8], circSOD2 [9], circEDEM2 [9]) a significant and distinct overexpression in RCC (e.g., >4-fold) and seemed, in our opinion, the most interesting biomarker candidates.

## 5. Conclusions

circRNAs in ccRCC tissue provide diagnostic and prognostic information: circEGLN3 expression levels allow discrimination of ccRCC and normal renal tissue, and circEHD2 and circNETO2 levels provide independent prognostic information regarding patients’ survival following nephrectomy. Thus, circRNAs could help to optimize individual treatment and ultimately improve ccRCC patients’ survival.

## Figures and Tables

**Figure 1 cancers-13-02177-f001:**
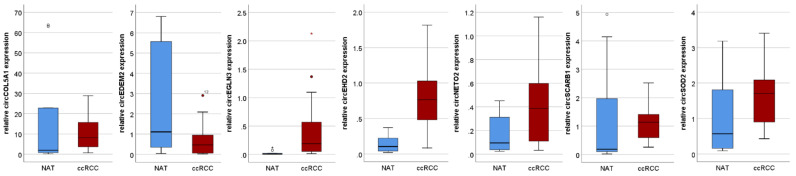
The expression of circEHD2, circEGLN3, and circNETO2 is increased in clear cell renal cell carcinoma (ccRCC) compared with normal adjacent renal (NAT) tissues, whereas expression levels of circSCARB1, circCOL5A1, and circSOD2 are not different in ccRCC and NAT. Circles indicate outliers.

**Figure 2 cancers-13-02177-f002:**
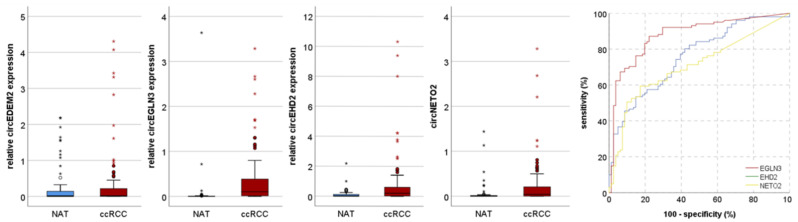
The expression of circEHD2 (*p* < 0.001), circEGLN3 (*p* < 0.001), and circNETO2 (*p* < 0.001) is increased in clear cell renal cell carcinoma (ccRCC) compared with normal renal (NAT) tissues. The Receiver Operator Characteristic analysis demonstrates that circEGLN3 allows discrimination of NAT and ccRCC with high accuracy. Circles indicate outliers, asterixis extreme values.

**Figure 3 cancers-13-02177-f003:**
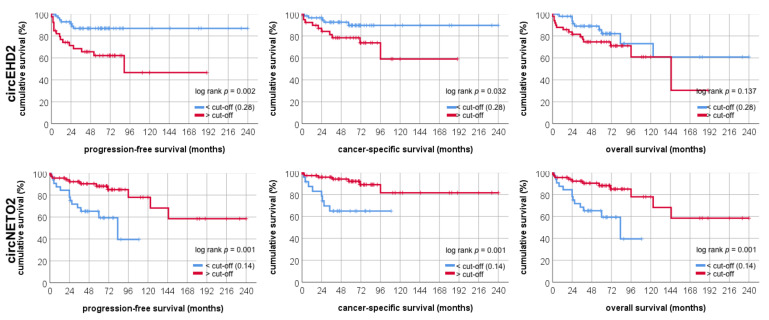
Kaplan–Meier estimates demonstrate that circRNA expression of circEHD2 and circNETO2 is associated with clear cell renal cell carcinoma patients’ survival following nephrectomy.

**Figure 4 cancers-13-02177-f004:**
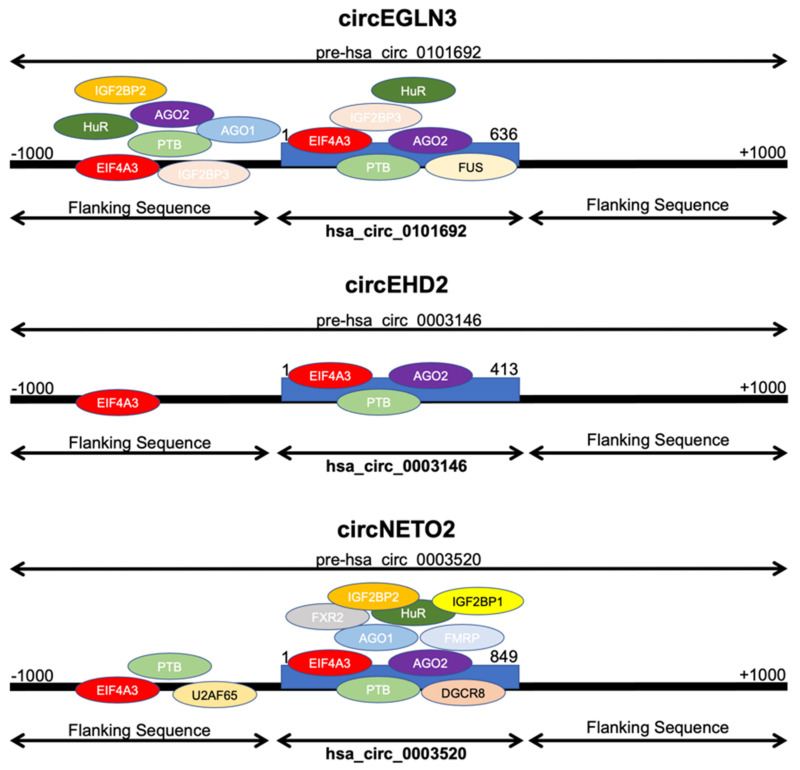
The circRNAs circEHD2, circEGLN3, and circNETO2 have several putative RNA-binding protein binding sites in circRNA junction and the 5′-flanking sites.

**Figure 5 cancers-13-02177-f005:**
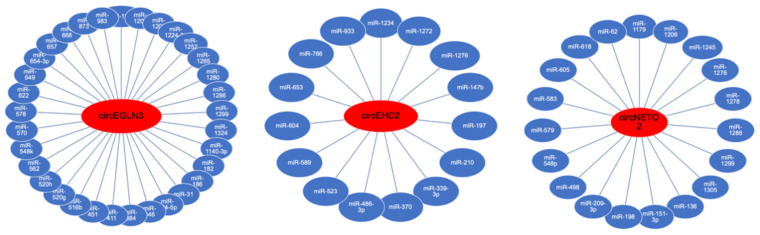
The circRNAs circEHD2, circEGLN3, and circNETO2 have multiple putative binding sites for miRNAs.

**Table 1 cancers-13-02177-t001:** Clinicopathological parameters of the study cohorts.

Clinical Parameter	Discovery Cohort		Validation Cohort	
	ccRCC (*n* = 20)	Normal (*n* = 10)	ccRCC (*n* = 101)	Normal (*n* = 81)
Sex				
male	15 (75.0%)	6 (60.0%)	65 (64.4%)	59 (72.8%)
female	5 (25.0%)	4 (40.0%)	36 (35.6%)	22 (27.2%)
Age				
mean	65.4	58.7	63.7	64.1
min-max	43–78	43–78	36–89	36–89
pT-stage				
pT1	10 (50.0%)		61 (60.4%)	
pT2	0 (0%)		11 (10.9%)	
pT3	9 (45.0%)		28 (27.7%)	
pT4	1 (5.0%)		1 (1.0%)	
lymph node metastasis				
cN0/pN0	17 (85.0%)		101 (100%)	
pN1	3 (15.0%)		0 (0%)	
distant metastasis				
cM0	10 (50.0%)		95 (94.1%)	
cM1	10 (50.0%)		6 (5.9%)	
Grading				
grade 1	1 (5.0%)		11 (10.9%)	
grade 2	12 (60.0%)		74 (73.3%)	
grade 3	3 (15.0%)		14 (13.9%)	
grade 4	4 (20.0%)		2 (2.0%)	

**Table 2 cancers-13-02177-t002:** Uni- and multivariate Cox regression analysis for progression-free survival.

Parameter	Univariate Analysis		Multivariate Analysis	
	*p*-Value	HR (95% CI)	*p*-Value	HR (95% CI)
Expression of circRNA				
Low (<cut-off)		1.00		1.00
High circEDEM2	0.246	1.68 (0.70–4.02)		
High circEGLN3	0.492	0.70 (0.26–1.92)		
High circEHD2	0.005	3.64 (1.48–8.92)	0.009	3.58 (1.37–9.38)
High circNETO2	0.002	0.26 (0.11–0.61)	0.001	0.17 (0.60–0.50)
Clinicopathological parameters				
Grading				
G1/2		1.00		1.00
G3/4	<0.001	5.40 (2.29–12.72)	<0.001	9.40 (3.38–26.09)
pT-stage				
pT1/2		1.00		1.00
pT3/4	0.002	3.76 (1.61–8.75)	0.009	3.32 (1.34–8.20)
cM-stage				
cM0		1.00		1.00
cM1	0.002	5.79 (1.94–17.25)	<0.001	11.91 (3.03–46.82)

**Table 3 cancers-13-02177-t003:** Uni- and multivariate Cox regression analysis for cancer-specific survival.

Parameter	Univariate Analysis		Multivariate Analysis	
	*p*-Value	HR (95% CI)	*p*-Value	HR (95% CI)
Expression of circRNA				
Low (<cut-off)		1.00		1.00
High circEDEM2	0.232	1.68 (7.16–3.97)		
High circEGLN3	0.883	0.938 (0.40–2.20)		
High circEHD2	0.005	3.64 (1.48–8.92)	0.042	2.67 (1.04–6.85)
High circNETO2	0.001	0.24 (0.10–0.56)	0.001	0.14 (0.05–0.43)
Clinicopathological parameters				
Grading				
G1/2		1.00		1.00
G3/4	<0.001	5.40 (2.29–12.72)	<0.001	11.66 (3.95–34.40)
pT-stage				
pT1/2		1.00		1.00
pT3/4	0.002	3.76 (1.61–8.75)	0.024	2.87 (1.15–7.17)
cM-stage				
cM0		1.00		1.00
cM1	0.002	5.78 (1.94–17.25)	<0.001	13.82 (3.386–56.39)

**Table 4 cancers-13-02177-t004:** Uni- and multivariate Cox regression analysis for overall survival.

Parameter	Univariate Analysis		Multivariate Analysis	
	*p*-Value	HR (95% CI)	*p*-Value	HR (95% CI)
Expression of circRNA				
Low (<cut-off)		1.00		1.00
High circEDEM2	0.541	1.30 (0.56–3.03)		
High circEGLN3	0.677	0.81 (0.30–2.21)		
High circEHD2	0.019	3.07 (1.20–7.85)	0.008	3.91 (1.43–10.67)
High circNETO2	0.010	0.32 (0.13–0.76)	0.001	0.15 (0.05–0.46)
Clinicopathological parameters				
Grading				
G1/2		1.00		1.00
G3/4	<0.001	5.40 (2.29–12.72)	<0.001	9.95 (3.56–27.85)
pT-stage				
pT1/2		1.00		1.00
pT3/4	0.002	3.76 (1.61–8.75)	0.007	3.52 (1.42–8.70)
cM-stage				
cM0		1.00		1.00
cM1	0.002	5.79 (1.94–17.25)	<0.001	14.03 (3.51–56.13)

## Data Availability

The raw PCR data are available on request from the corresponding author.

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
