# Peer review of "CircEHD2, CircNETO2 and CircEGLN3 as Diagnostic and Prognostic Biomarkers for Patients with Renal Cell Carcinoma"

_cancers, 2021, doi:10.3390/cancers13092177_

Round 1

Reviewer 1 Report

This research article by Frey et al. aimed at quantifying circular RNAs (circRNAs) in tissue specimens from clear cell renal carcinoma patients and normal tissues. Seven circRNAs were selected for investigation of their expression profile, three of which showed putative biomarker utility, namely circEHD2, circENGLN3, and circNETO2. The study included a sufficient sample cohort, larger from previous relevant studies; however, there are several major issues in the Manuscript. First of all, the Title seems misleading, and is more suitable for a Review article rather than a research study. Additionally, the Introduction lacks important information, while there are also some inconsistencies (e.g. circRNAs are mentioned as non-coding RNAs, and at the same time they translate proteins). Regarding the methodology, the overall approach is poor of experiments and, generally, lacks novelty and originality. In the Results, the expression levels of circSCARB1, circCOL5A1, and circSOD2 were not found to be different between tumor and normal renal samples, whereas in the Introduction Authors state that all seven circRNAs were found upregulated in previous studies, leading to contradiction. Moreover, the Figures provided should be resized and more clearly presented since currently they are hard to read and comprehend. The Discussion is also lacking in content, and is more of a repetition of the Results rather than an in-depth commentary of the significance of the results, and the literature needs to be expanded as well. Finally, potential limitations of this study are not addressed by the Authors.

Author Response

This research article by Frey et al. aimed at quantifying circular RNAs (circRNAs) in tissue specimens from clear cell renal carcinoma patients and normal tissues. Seven circRNAs were selected for investigation of their expression profile, three of which showed putative biomarker utility, namely circEHD2, circENGLN3, and circNETO2. The study included a sufficient sample cohort, larger from previous relevant studies; however, there are several major issues in the Manuscript.

First of all, the Title seems misleading, and is more suitable for a Review article rather than a research study.

As suggested by the reviewer, we have modified the title of our study.

Additionally, the Introduction lacks important information, while there are also some inconsistencies (e.g., circRNAs are mentioned as non-coding RNAs, and at the same time they translate proteins). Regarding the methodology, the overall approach is poor of experiments and, generally, lacks novelty and originality.

We have revised the introduction section.

As stated, our study was designed to validate circRNA dysregulation. At present, quantification of circRNAs using quantitative real-time PCR after RNase R treatment (to degrade linear RNA) is the gold-standard for circRNA quantification. The circRNA candidates were chosen based on previous reports.

In the Results, the expression levels of circSCARB1, circCOL5A1, and circSOD2 were not found to be different between tumor and normal renal samples, whereas in the Introduction Authors state that all seven circRNAs were found upregulated in previous studies, leading to contradiction. Moreover, the Figures provided should be resized and more clearly presented since currently they are hard to read and comprehend.

Previous studies investigated circRNAs in small cohorts, and most studies only performed microarray analyses. Thus, validation studies are required to allow a better understanding of its role as biomarker. Therefore, it is an important finding that circSCARB1, circCOL5A1, and circSOD2 were not overexpressed in our large validation cohort. 

All Figures have been revised to improve readability.

The Discussion is also lacking in content, and is more of a repetition of the Results rather than an in-depth commentary of the significance of the results, and the literature needs to be expanded as well. Finally, potential limitations of this study are not addressed by the Authors.

As suggested, we have improved the discussion and added a section on the limitations of our study.

Reviewer 2 Report

Frey et.al., have proposed the diagnostic/prognostic roles of three circularRNAs in RCC. Building on previously published analysis of circularRNAs in RCC tissue samples, they have tested seven circRNAs; and found one to be of diagnostic significance and other two with prognostic significance.

The Cohort selection, sample preparation, experiments and statistical analysis were done right. The large size cohort supports the conclusion.

However, few minor points need to be addressed.

  1. Line 169, low circNETO2 levels were predictive of progression-free survival to predictive of decreased progression-free survival can be clearer, as the previous line 168 explicitly states shortened progression free survival (for circEHD2).
  2. The authors can discuss the sensitivity of circEHD2 and circNETO2 in the Discusssion; as the ROC analysis revealed that these two circRNAs were not that sensitive in validation cohort.

Author Response

Frey et.al., have proposed the diagnostic/prognostic roles of three circularRNAs in RCC. Building on previously published analysis of circularRNAs in RCC tissue samples, they have tested seven circRNAs; and found one to be of diagnostic significance and other two with prognostic significance.

The cohort selection, sample preparation, experiments and statistical analysis were done right. The large size cohort supports the conclusion. However, few minor points need to be addressed.

  1. Line 169, low circNETO2 levels were predictive of progression-free survival to predictive of decreasedprogression-free survival can be clearer, as the previous line 168 explicitly states shortened progression free survival (for circEHD2).

We have modified the sentence as suggested by the reviewer.

  1. The authors can discuss the sensitivity of circEHD2 and circNETO2 in the Discusssion; as the ROC analysis revealed that these two circRNAs were not that sensitive in validation cohort.

As suggested by the reviewer, we have discussed the diagnostic accuracy of circREHD2 and circNETO2 more comprehensively.

Reviewer 3 Report

This study by Frey et al investigate 7 circular RNAs as prognostic factors for renal cell carcinoma. A strength of the study is the fresh-frozen series of samples used, which was divided into a discovery and validation set. A weak point is the limited number of circRNA analyzed, and differences between normal and tumor tissue (Figs 1 and 2) have previously been described. In addition to this limitation in the novelty of the study, there are additional issues that should be addressed:

  • Did the selection of the 7 RNAs follow and standardized procedure? If this is the case, please, explain thresholds applied.
  • In the Introduction the authors indicate that circRNA have higher stability than mRNA, which could be interesting for a potential biomarker. However, a recent publication suggests the opposite (PMID: 32802191). Please, comment.
  • There is a lack of metastatic cases. The implications for the study should be commented in the Discussion
  • Why is the expression dichotomized into high and low expressors in the survival analysis? While this is needed for a Kaplan-Meier analysis, this is not so for cox-regression. Please provide these analyses and also include in Table 2 the prognostic risk group of the patients.
  • Statistical methods should be expanded to provide detail of approaches applied. Provide also with cut-off values and the rationale for them.

Author Response

This study by Frey et al investigate 7 circular RNAs as prognostic factors for renal cell carcinoma. A strength of the study is the fresh-frozen series of samples used, which was divided into a discovery and validation set. A weak point is the limited number of circRNA analyzed, and differences between normal and tumor tissue (Figs 1 and 2) have previously been described.

In addition to this limitation in the novelty of the study, there are additional issues that should be addressed:

- Did the selection of the 7 RNAs follow and standardized procedure? If this is the case, please, explain thresholds applied.

As suggested, we have described the selection process of the circRNAs more detailed in the methods section. The selection of circRNAs was based on an at least 4-fold overexpression and statistical significance in former experiments. Furthermore, successful primer design was mandatory for candidate circRNAs.

- In the Introduction the authors indicate that circRNA have higher stability than mRNA, which could be interesting for a potential biomarker. However, a recent publication suggests the opposite (PMID: 32802191). Please, comment.

We have addressed this remark in the discussion section.

- There is a lack of metastatic cases. The implications for the study should be commented in the Discussion

Our study included only few patients with distant metastasis and only 3 patients in the discovery cohort had lymph node metastases. In our clinic, cytoreductive nephrectomy of patients with RCC was only rarely performed in the past years as recent studies question the benefit of nephrectomy in patients with metastatic RCC, and thus most of the samples in the biobank are from patients with localized disease. The discovery cohort included each 10 patients with localized and metastatic RCC, whereas the selection of samples for the validation cohort was randomly from the biobank. We explained this in the Methods section, and discuss this limitation.

- Why is the expression dichotomized into high and low expressors in the survival analysis? While this is needed for a Kaplan-Meier analysis, this is not so for cox-regression. Please provide these analyses and also include in Table 2 the prognostic risk group of the patients.

We agree with the reviewer that dichotomization of continuous data is a debatable issue. in our opinion, dichotomization makes the presentation and interpretation of results easier and we therefore decided to use the dichotomized circRNA expression levels for cox regression analysis.

We did not include prognostic risk groups (e.g., Motzer score or Heng criteria) for patients with metastatic disease as the number of patients with metastatic disease was small in the validation cohort (n=6). This is also discussed as limitation.

- Statistical methods should be expanded to provide detail of approaches applied. Provide also with cut-off values and the rationale for them.

As described in the Methods section, an optimal cut-off for discrimination of patients with/without recurrence based on circRNA expression was determined using ROC analyses and the Youden index. The cut-off values are included in Figure 3.

Round 2

Reviewer 1 Report

The revised manuscript has been improved. However, there are still several major issues that need to be addressed, as explained below, before this manuscript becomes acceptable for publication in in “Cancers”:

  • The authors are strongly advised to describe putative interactions of the studied circRNAs with microRNAs and RNA-binding proteins (RBPs). There are multiple bioinformatical tools that could be used for this purpose. For instance, the authors may have a look at the paper of Karousi et al. (IJMS, 2020), in which such predictions have successfully been used.
  • A Figure showing the exon structure of the studied circRNAs as well as the position of PCR primers used would be very helpful for the readers.
  • It is becoming more and more evident that multiple alternative circRNAs may result from the same primary transcript. The authors should comment on that. Moreover, are the primer pairs specific for a single circRNA of each gene? Or do they amplify multiple alternative circRNAs? Although the Authors refer to previously published papers for these sets of primers, it should be taken into consideration that knowledge about alternatively spliced, linear and circular RNA transcripts is continuously being updated.
  • Did the authors check the prerequisites for qPCR-based quantification in validation experiments? These data should be provided, at least as Supplemental Material.
  • It would make sense to provide a Supplementary Table with all primer sets and expected amplicon sizes.
  • Which calibrator and reference gene(s) were used in qPCR? This information needs to be clearly reported in Materials and Methods.
  • The best cutoff values based on ROC analysis are optimal for diagnostic purposes. However, there are other ways to determine the optimal prognostic cutoff; for instance, please see X-tile (Camp et al., Clin. Cancer Res., 2004). The authors are advised to use this method (or another appropriate method) to determine the optimal prognostic cutoff for the expression values of each circRNA.
  • Minor English grammar and syntax errors need to be corrected.

Author Response

The revised manuscript has been improved. However, there are still several major issues that need to be addressed, as explained below, before this manuscript becomes acceptable for publication in in “Cancers”:

  • The authors are strongly advised to describe putative interactions of the studied circRNAs with microRNAs and RNA-binding proteins (RBPs). There are multiple bioinformatical tools that could be used for this purpose. For instance, the authors may have a look at the paper of Karousi et al. (IJMS, 2020), in which such predictions have successfully been used.

We have performed additional bioinformatic analyses using the CircInteractome tool. Figure 4 and Figure 5 provides information on RBP and miRNA interactions with the most interesting circRNAs (circEGLN3, circED2, circNETO2) in our ccRCC cohort. Please see also the added section at the end of the Discussion.

  • A Figure showing the exon structure of the studied circRNAs as well as the position of PCR primers used would be very helpful for the readers.

In order to provide more information for the reader, we have added the requested information in the Supplementary Information for those circRNAs with newly designed PCR primers.

  • It is becoming more and more evident that multiple alternative circRNAs may result from the same primary transcript. The authors should comment on that. Moreover, are the primer pairs specific for a single circRNA of each gene? Or do they amplify multiple alternative circRNAs? Although the Authors refer to previously published papers for these sets of primers, it should be taken into consideration that knowledge about alternatively spliced, linear and circular RNA transcripts is continuously being updated.

All primers were specific for the specified circRNA splice variant; the specific splice variant is mentioned in the Supplementary Information.

  • Did the authors check the prerequisites for qPCR-based quantification in validation experiments? These data should be provided, at least as Supplemental Material.

A MIQE-compliant report is provided in the Supplementary Information.

  • It would make sense to provide a Supplementary Table with all primer sets and expected amplicon sizes.

The requested information is also provided in the Supplementary Information.

  • Which calibrator and reference gene(s) were used in qPCR? This information needs to be clearly reported in Materials and Methods.

We used cel-miR-39 as reference gene for the qPCR.

  • The best cutoff values based on ROC analysis are optimal for diagnostic purposes. However, there are other ways to determine the optimal prognostic cutoff; for instance, please see X-tile (Camp et al., Clin. Cancer Res., 2004). The authors are advised to use this method (or another appropriate method) to determine the optimal prognostic cutoff for the expression values of each circRNA.

We now employed Cutoff Finder (Budczies et al. PLoS One 2012; https://molpathoheidelberg.shinyapps.io/CutoffFinder_v1/) to determine the optimal threshold for Kaplan Meier analyses.

  • Minor English grammar and syntax errors need to be corrected.

Grammar has been thoroughly revised.

Reviewer 3 Report

The authors do not improve the quality of the manuscript sufficiently nor address all issues raised in the revision procedure.

Author Response

The authors do not improve the quality of the manuscript sufficiently nor address all issues raised in the revision procedure.

Please find below comments to your first remarks. We have addressed your comments more detailed in the second revision.

  • Did the selection of the 7 RNAs follow and standardized procedure? If this is the case, please, explain thresholds applied.

We systematically reviewed the literature on circRNAs in renal cell carcinoma. Several microarray studies (references 8-11) identified circRNAs with distinct overexpression (> 4-fold) in RCC tissue, however the cohorts studied were small, and validation experiments in an enlarged cohort using a different technique (PCR) were reported only for selected circRNAs. We therefore selected circRNAs which have not been validated in former studies. This procedure is explained more detailed in the second paragraph of the Introduction section.

  • In the Introduction the authors indicate that circRNA have higher stability than mRNA, which could be interesting for a potential biomarker. However, a recent publication suggests the opposite (PMID: 32802191). Please, comment.

The reviewer refers to the study of Rochow et al. which observed that circRNA expression may be degraded similar as mRNA. However, in clinical samples with a high RNA integrity (RIN > 6) the study did not observe a significant degradation of circRNAs. We only used fresh-frozen tissues in order to guarantee a high RNA integrity, and we therefore do not expect a bias in the circRNA expression. We discuss the issue in the first section of the Discussion.

  • There is a lack of metastatic cases. The implications for the study should be commented in the Discussion

Our validation cohort lacks unfortunately samples from patients with metastatic disease; this is owed the fact that samples were randomly selected from the biobank without knowledge of the clinical information. The number of samples from patients with metastatic disease is limited in our biobank, especially since the CARMENA trial indicated improved outcome in patients undergoing sunitinib therapy without cytoreductive nephrectomy which changed our clinical routine and patients with metastatic disease no longer undergo cytoreductive surgery.

The lack of metastatic cases in the validation cohort is on the other hand a strength of the study: the survival analyses indicate a prognostic value of circRNA expression in patients with localized/locally advanced disease undergoing nephrectomy. Thus, the analysis of circRNA expression could aid the clinician to improve follow-up in patients after surgery with curative intention, and allow identification of patients at increased risk of RCC recurrence. Furthermore, these patients could be of special interest for adjuvant tumor therapy. Please see the last paragraph in the discussion section.

  • Why is the expression dichotomized into high and low expressors in the survival analysis? While this is needed for a Kaplan-Meier analysis, this is not so for cox-regression. Please provide these analyses and also include in Table 2 the prognostic risk group of the patients.

Cutoff Finder (Budczies et al. PLoS One 2012) was used to determine optimal cutoffs for Kaplan Meier analyses. Using dichotomized variables for Cox regression analyses is commonly in the literature. We also assessed circRNA expression as continuous variable, however, the results did not reach statistical significance.

As mentioned in the discussion of the study limitations, we did not include prognostic risk groups (i.e., Heng or Motzer criteria) as we did not study patients with metastatic disease in the validation cohort.

  • Statistical methods should be expanded to provide detail of approaches applied. Provide also with cut-off values and the rationale for them.

As mentioned above, we used Cutoff Finder (Budczies et al. PLoS One 2012) in the revised version of our manuscript to determine optimal cutoffs for survival analyses.

Round 3

Reviewer 1 Report

The issues raised by the Reviewers were properly addressed.

Reviewer 3 Report

The authors have answered satisfactorily the issues raised.